

Variation characteristics of sporadic-E layer in East Asia
based on long-term data
**Hai-Sheng Zhao** [1]**, Jie Feng*** [2]**, Yang Liu** [2]**, Zheng-Wen Xu** [2]**, Jian Wu** [3]**, Kun Xue** [2]**,**
**Huai-Yun Peng** [2]**, Zing-Hua Ding** [3]
1 School of Electronic Science and Technology, Hainan University, Haikou 570228, China
2 National Key Laboratory of Electromagnetic Environment, China Research Institute of Radiowave
Propagation, Qingdao 266107, China
3 Kunming Electro-Magnetic Environment National Observation and Research Station, China
Research Institute of Radiowave Propagation, Qujing 655500, China.
*    Correspondence: fengjie@crirp.ac.cn
**Abstract**:The characteristics of ionospheric intensity, spatial distribution, diurnal variation,
seasonal variation and long-term variation in East Asia are studied by using the ionospheric
observation data from 21 ionosonde stations in China and Japan over the past 60 years. It is found
that the Es layer intensity in East Asia is much higher than the global average, and the intensity
center is located near the 30° N line, and weakens to low and high latitudes. At the same time, the
intensity center of Es layer is not fixed, and the intensity center migrates with diurnal and seasonal
variations. The regions with stronger Es showed a long-term downward trend, while the regions
with weaker Es showed a long-term upward trend in East Asia.
**Key words** East Asia, Sporadic E Layer, Variation Characteristics, Long-term Trend, Spatial
Distribution

## 1 Introduction

Sporadic E (Es) is a transient electron density enhancement structure that occurs at altitude of
90-140 kilometers and can significantly affect the propagation of radio waves. The Es layer may
occur during the day or at night, and its variations with latitude and time are pronounced." The Es
layer is a special structure within the ionosphere, unlike the regular E layer that exhibits stable and
regular morphological structures and trends. Instead, it is a transient and irregular strong
ionization layer, with a height range of 90 to 140 km and a thickness ranging from a few hundred
meters to 1 km [Danilov et al., 2020; Pignalberi et al., 2014]. Its horizontal scale varies from tens
of kilometers to hundreds of kilometers, and it drifts at speeds ranging from 20 to 300 m/s [Maeda
et al., 2016]. The seasonal distribution of the Es layer in the ionosphere is uneven, with higher
occurrence frequencies in summer months from May to August and lower frequencies in other



months [Sivakandan et al., 2023; Jacobi et al., 2019;  Haldoupis et al., 2007]. The Es layer in the
ionosphere exhibits significant diurnal variation, with higher occurrence frequencies during the
day and lower frequencies during the night. The electron density in the Es layer of the ionosphere
is exceptionally high, reaching up to 100 times the electron density of the regular E layer.
Therefore, the Es layer in the ionosphere is capable of reflecting radio waves that would otherwise
penetrate through to the F layer, resulting in the reflection and scattering of HF/VHF frequency
radio waves. The maximum single-hop propagation distance can exceed 2000 km.
In the early 20th century, unexpected reflected signals received by instruments such as
altimeters, television, and amplitude-modulated radios sparked great interest among researchers
[Whitehead,1970, 1989], leading to the beginning of studies on the Es layer of the ionosphere.
Since the 1960s, scientists have gradually gained understanding of the Es layer and its related
characteristics through observations and analysis using instruments such as ionospheric sounders
[Whitehead,1970, 1989; Reddy et al., 1968], incoherent scatter radars [Swartz et al., 1974;
Ioannidis et al., 1972], sounding rockets [Yamamoto et al., 1998; Pfaff et al., 1998; Kelley et al.,
1995; Smith, 1970; Seddon et al., 1962], coherent scatter radars, and other methods [Haldoupis et
al., 1996, 1997]. They have established and continuously improved theories on the formation
mechanism of the Es layer in the ionosphere, and have gradually acquired knowledge about its
radio wave propagation characteristics. The wind shear theory is considered to be the primary
mechanism for the formation of the mid-latitude Es layer [Axford et al., 1963; Didebulidze et al.,
2015]. Under the influence of the geomagnetic field, the horizontal wind generates vertical shear
force on ions at the height of the ionospheric dynamo layer, compressing the ion constituents and
forming a thin layer of high-density ionization, namely the Es layer. Tidal waves, planetary waves,
and gravity waves affect the wind shear, causing metallic ions and molecular ions to move and
converge, forming thin high-density plasma layers in mid-latitudes [Qiu et al., 2023; Haldoupis et
al., 2006; Axford et al., 1966; Helmboldt et al., 2016; Christos et al., 2004, 2006; Davis et al.,
2006; Tepley et al., 1985; Macleod et al., 1975]. Although the wind shear theory is currently the
mainstream explanation for the formation of the Es layer, the theory itself is still not fully
developed [Liu et al., 2022], and there are still difficulties in explaining certain phenomena, such
as the summer anomalies of the mid-latitude Es layer and the extremely uneven distribution of Es
layer intensity globally.
The unpredictability and highly uneven spatial-temporal distribution of the Es layer have
sparked great interest among researchers, initiating relentless studies for over half a century. In the
1950s, Smith [Smith et al., 1957] proposed the concept of the "Far East Anomaly" based on
statistical analysis of global vertical sounding records and research on VHF over-the-horizon
propagation phenomena. The Far East Anomaly refers to the phenomenon where the occurrence



rate and intensity of Es layers in the mid-latitude regions of the Far East are exceptionally high far
exceeding those in other regions at the same latitude. Correspondingly, during the summer in the
southern hemisphere, the Es layer in the South American region is also relatively strong with a
higher occurrence rate, but it is not as prominent as the Far East Anomaly. With the development
of satellite technology, the method of detecting the spatial distribution of the Es layer using
satellite beacons has gradually matured. The distribution of the global Es layer has been obtained
based on GPS radio occultation technique [Arras et al., 2009], which has greatly expanded the
research methods for the Es layer and has epoch-making significance. Due to the scarcity of early
ionosonde stations for the Es layer and insufficient data accumulation, early studies on the
morphology of the Es layer could only provide a rough global distribution of the Es layer. The
radio occultation observations also has its limitations, such as it cannot continuously observe the
Es layer at fixed location compared to the ground-based observations. It can only estimate Es layer
intensity through phase and amplitude variations, and its accuracy needs to be improved.
With the increasing number of global ionosonde stations and the accumulation of data over
time, conducting long-term studies on the Es layer's characteristics using global ionosonde data is
of significant scientific importance. Eastern Asia is located in the peak region of the Far East
Anomaly, making it uniquely advantageous for research. In this study, utilizing over 60 years of
Es layer observation data from 21 ionosonde stations in China and Japan [Zhao, 2024a], we
conducted in-depth research on the intensity characteristics, spatial distribution, diurnal variation,
seasonal variation, and long-term variations of the Es layer in the Eastern Asian. The data analysis
software on which this article is based are available in Zhao. [2024b]. The research findings of this
study are of great significance for exploring the formation mechanisms of the Es layer, analyzing
the spatial-temporal distribution and long-term trends of Es layer intensity. Additionally, the Es
layer exhibits high scattering efficiency for VHF frequency signals and has advantages in VHF
over-the-horizon propagation. This makes it valuable for achieving long-distance, sudden
communication and meeting the minimum requirements for information transmission.

## 96 2 Ionosonde Station Network and Data Sources in China and Japan

Ground-based radio vertical sounding of the ionosphere is one of the fundamental methods
for ionospheric exploration and was the only effective means of investigation before the advent of
rockets and artificial satellites. After years of development, the network of ionosonde stations in
China has gradually covered a vast area of our country, spanning 30 degrees in geomagnetic
latitude and longitude. The longitude intervals are between 3 to 10 degrees, while the latitude
intervals range from 3 to 6 degrees. Particularly, in the geomagnetic sector of 180 to 193 degrees





east, a meridional chain of six stations has been established, forming a well-designed network of
ionosonde stations that meets the needs of shortwave communication frequency prediction and
related radio wave propagation studies. Currently, through the joint efforts of various universities
and research institutions in China, significant achievements have been made in the field of
ionospheric exploration. These include shortwave communication frequency prediction,
ionospheric disturbance forecasting, and the refractive correction for tracking and locating rocket
experiments.

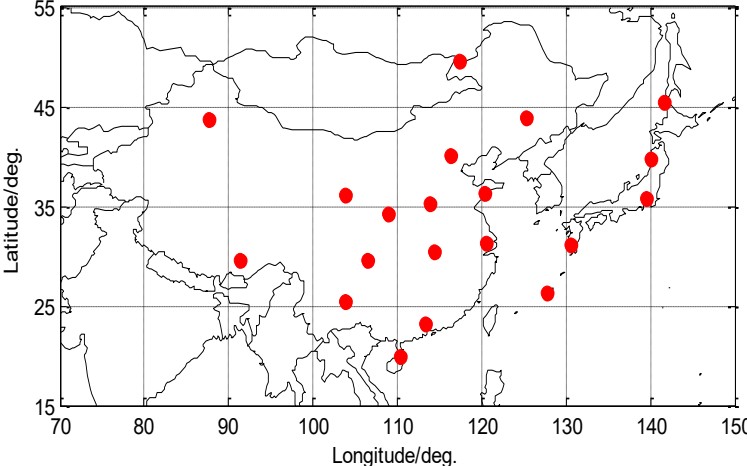

Fig.1 ionosonde stations in China and Japan

The Es data primarily comes from ionosonde stations in China, including Beijing, Changchun,

Chongqing, Guangzhou, Hainan, Lhasa, Manzhouli, Urumqi, Wuhan, Xinxiang, Kunming,
Qingdao, Suzhou, Sheshan, Xi'an, as well as surrounding areas in China. Additionally, data from
the OKINAWA (Okinawa Island, Japan), YAMAGAWA (Yamagawa Prefecture, Japan), Koku,
Akita, and Wakkanai ionosonde stations are also included, as shown in Figure 1. The names,
nationalities, coordinates, and data periods of each stations are list in Table 1.

Table 1 the observation stations and terms of Es

| Index | Station name | Country | Longitude | Latitude | Time period |
|-------|--------------|---------|-----------|----------|-------------|
| 1 | Beijing | China | N40.11° | E116.27° | 1958~2020 |
| 2 | Changchun | China | N43.84° | E125.27° | 1957~2020 |
| 3 | Chongqing | China | N29.50° | E106.40° | 1958~2020 |
| 4 | Guangzhou | China | N23.15° | E113.35° | 1958~2020 |
| 5 | Haikou | China | N20.00° | E110.33° | 1958~2020 |
| 6 | Lanzhou | China | N36.06° | E103.87° | 1958~2020 |
| 7 | Lhasa | China | N29.63° | E91.28° | 1970~2020 |
| 8 | Manzhouli | China | N49.58° | E117.45° | 1958~2020 |
| 9 | Urumchi | China | N43.75° | E87.63° | 1958~2020 |
| 10 | Qingdao | China | N36.24° | E120.41 | 2000~2020 |
| 11 | Sheshan | China | N31.00° | E121.24° | 1961~1966 |
| 12 | Kunming | China | N 25.50° | E103.80° | 2007~2020 |



| 13 | Xinxiang | China | N35.30° | E113.95° | 2008~2020 |
|----|----------|-------|---------|----------|-----------|
| 14 | Suzhou | China | N31.30° | E120.65° | 2008~2020 |
| 15 | Xian | China | N34.23° | E108.92° | 2011~2020 |
| 16 | Wuhan | China | N30.50° | E114.40° | 1957~1998 |
| 17 | Akita | Japan | N39.70° | E140.10° | 1965~1993 |
| 18 | Okinawa | Japan | N26.30° | E127.80° | 1972~2010 |
| 19 | Yamagawa | Japan | N31.20° | E130.60° | 1965~2010 |
| 20 | Wakkanai | Japan | N45.40° | E141.70° | 1957~2005 |
| 21 | Koku | Japan | N35.70° | E139.50° | 1958~2005 |


## 3 Characteristics of Es layer intensity in East Asia

The monthly median of the Es layer critical frequency (foEs) is an important parameter for
assessing the Es layer intensity in a specific region. The foEs monthly median reflects the average
level of the Es layer intensity in that region and possesses a high level of reliability. Figure 2
presents the variations of the monthly median foEs with local time and month for 20 stations in
East Asia.

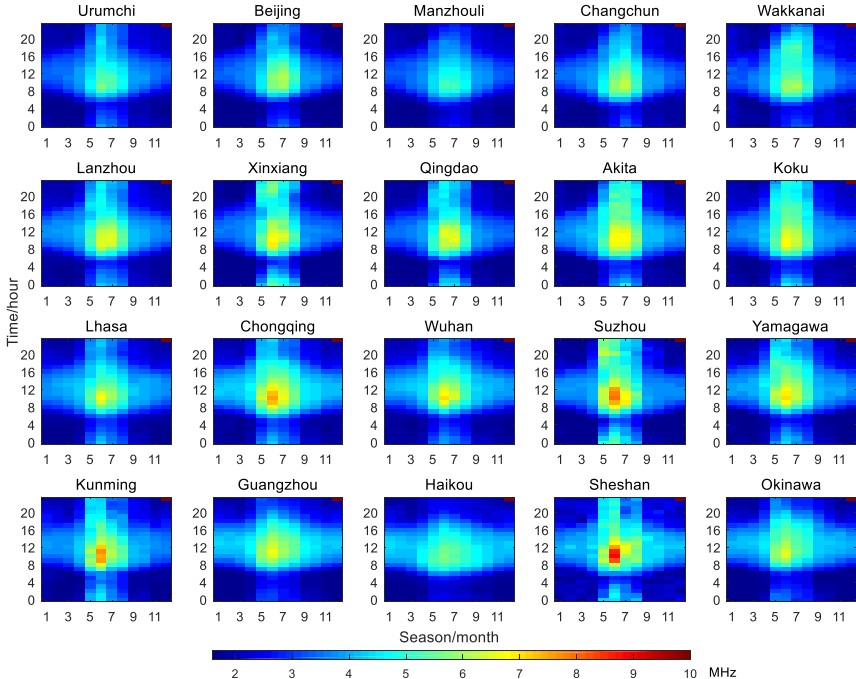


Fig.2 local time-month distributios of foEs monthly median located at different stations
According to figure 2, the foEs layer intensity during the summer months (May to August) is
significantly higher than in other seasons. The foEs layer intensity is also notably higher around



local noon compared to other times of the day. Additionally, the Es layer intensity in East Asia
exhibits a strong spatial non-uniformity. Generally, the distribution of Es layer intensity centers
around the latitude of 30 degrees north, decreasing towards lower and higher latitudes. The overall
intensity is higher at lower latitudes compared to higher latitudes, and the intensity is slightly
stronger in the eastern region compared to the western region. The maximum monthly average
foEs values for all stations are above 5 MHz, with some stations reaching even higher values
exceeding 9 MHz, which is significantly higher than the global average level [Smith et al., 1970].
## 4 Spatial distribution characteristics of Es layer in East Asia
The spatial distribution of foEs is crucial for investigating the characteristics of the Es layer
in a specific region. In this section, the distribution of the annual mean value of foEs in East Asia
is given by the Kriging interpolation method [Zhu et al., 20223; Liu et al., 2022] based on the
annual mean value of foEs at each station (as shown in figure 3).

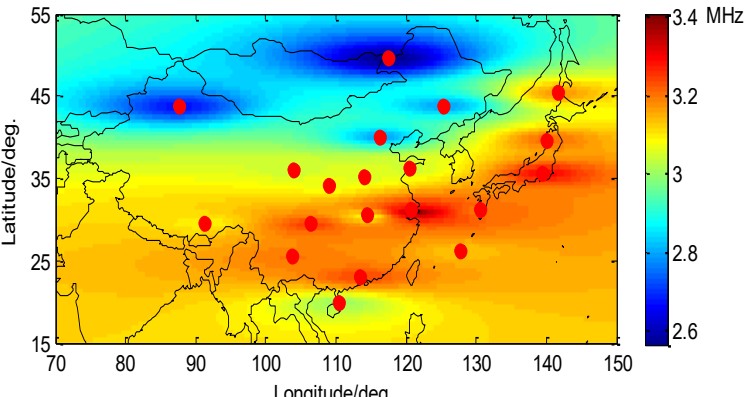


Fig.3 the distribution of foEs average values in East Asia
From Figure 3, it is shown that the Es layer intensity in East Asia exhibits a zonal distribution
along the latitude. The peak intensity of foEs occurs near 30 degrees north latitude. Over the years,
there has been ongoing debate regarding the center of global Es layer intensity. Some researchers
argue that the center of global Es layer intensity is near Wakkanai, Japan [Smith et al., 1970].
However, figure 3 clearly shows that while the average Es layer intensity in the sea area near
Wakkanai, Japan is relatively high, it is not the area with the highest intensity of Es layer. The
actual center of Es layer intensity should be located near Suzhou, China. Rather than considering
the Es layer intensity center as a single point, it is more appropriate to view it as a zonal region,
with the center of this region lying along the 30 degrees north latitude line.




# 5 Temporal distribution characteristics of Es layer in East Asia

## 5. 1 Diurnal variation characteristics

In order to further investigate the diurnal variation patterns of Es layer intensity in East Asia, the study presents the average variations of monthly median foEs values with time. Due to the large number of stations, five representative stations near 30 degrees north latitude, namely Lhasa, Chongqing, Wuhan, Suzhou, and Yamagawa, as well as six representative stations near 120 degrees east longitude, namely Manzhouli, Changchun, Qingdao, Suzhou, Guangzhou, and Haikou, were selected. The diurnal variation curves for each of these stations are provided in figure 4.

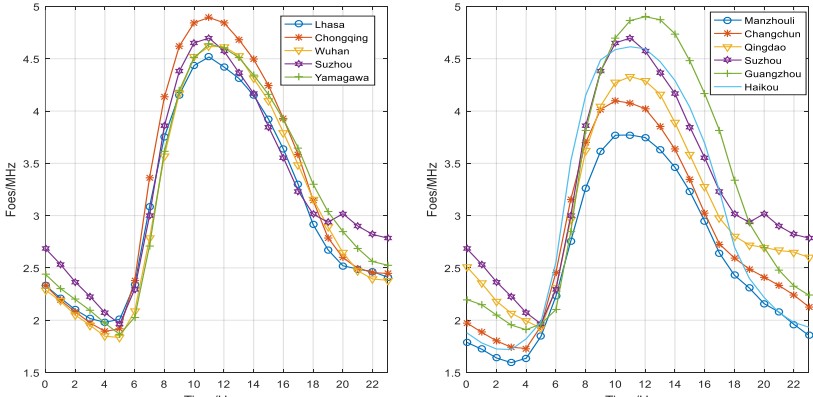

Fig.4 Diurnal variation curves of foEs monthly median

From figure 4, it is shown that the maximum values of foEs in East Asia generally occur around 11 AM, while the minimum values typically occur around 5 AM. At daytime, foEs values are significantly higher than during the nighttime.

To further investigate the diurnal variation characteristics of Es layer intensity in East Asia, figure 5 presents the spatial distribution characteristics of foEs monthly median values during daytime and nighttime. (Daytime is defined as 8 AM to 5 PM, and nighttime is defined as 9 PM to 6 AM the following day).

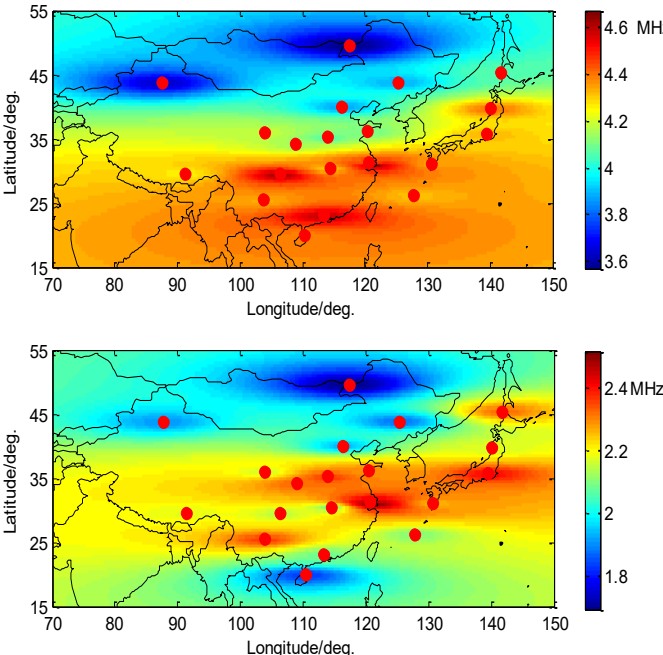

Fig.5 the day and night comparison of foEs average values

Figure 5 shows that at daytime, the center of Es layer intensity in East Asia is located in areas

such as Chongqing, Guangzhou, and Suzhou in China. During the nighttime, the center of Es layer
intensity migrates towards the northeast, with the strongest area appearing in regions such as
Suzhou and Qingdao in China, as well as Koku and Yamagawa in Japan. The diurnal drift in the
center of Es layer may be affected by environmental factors such as the diurnal variations of
background atmoshphere and climate.
5. 2 Seasonal variation characteristics

To further investigate the seasonal variation characteristics of the Es layer in East Asia, figure

6 presents the average variations of monthly median foEs values with seasons. The station
selection is the same as in section 5.1.





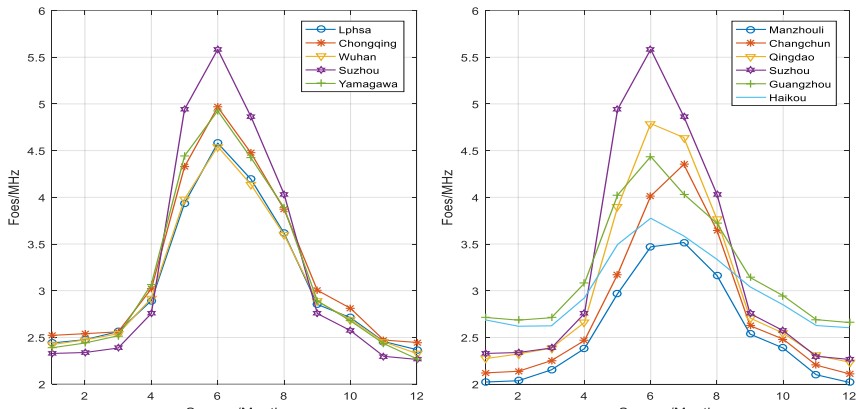


Fig.6 seasonal variation curves of foEs average values

According to figure 6, the maximum values of foEs in East Asia generally occur in June,

while the minimum values typically occur in December. The foEs values in summer are
significantly higher than in winter.

To further investigate the seasonal variation characteristics of Es layer intensity in East Asia,

figure 7 presents the spatial distribution characteristics of the average monthly median foEs values
during summer and winter. (Summer is defined as May to August, and winter is defined as
November to February of the following year).

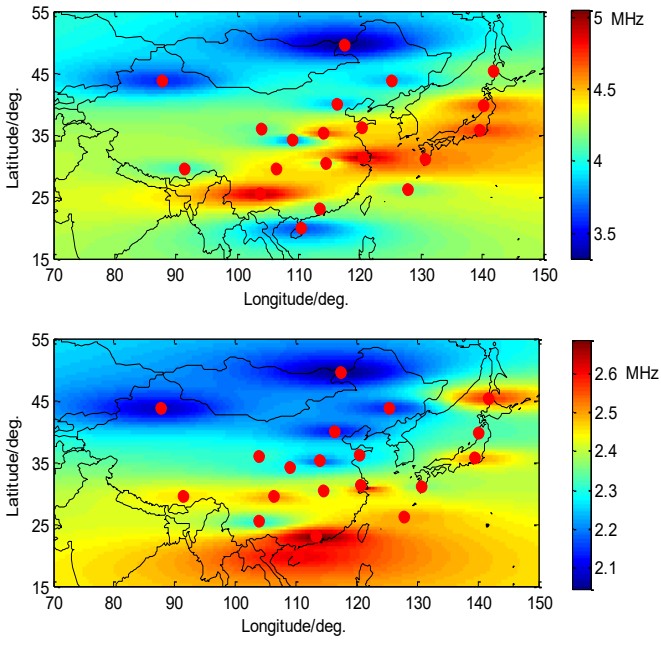


Fig.7 the summer and winter comparison of foEs average values





Figure 7 shows that during the summer in East Asia, the center of Es layer intensity is located
near 30 degrees north latitude, exhibiting a zonal distribution. However, during the winter, the
center of Es layer intensity migrates southward towards the Guangzhou and Haikou.
The proposal of the wind shear theory and the discovery of metallic ions provide a reasonable
explanation for the formation process of the Es layer. Measurements of ion density and wind
velocity through rocket experiments have confirmed the fundamental concept of wind shear
compression. However, the occurrence of anomalous phenomena in the mid-latitude Es layer
during summer poses a challenge to the wind shear theory. In order to address this issue, some
scholars have conducted in-depth research by linking the occurrence rate of mid-latitude Es layer
with planetary waves. They propose that planetary waves are also an important factor influencing
the mid-latitude Es layer and suggest that the viewpoint of planetary waves can provide a
reasonable explanation for the summer anomaly phenomenon. Furthermore, they indicate that
planetary waves modulate tidal amplitudes, load information onto tides, and indirectly affect the
Es layer through tides. They also predict that the modulation of tides by planetary waves is
achieved through nonlinear interference [Xu et al., 2022].
From the analysis of the probability distribution, intensity distribution, diurnal variation, and
seasonal variation of the Es layer, we have observed a general pattern: the center of Es layer
intensity seems to be chasing the center of high temperatures in the lower atmosphere. Regions
with higher average temperatures tend to exhibit stronger Es layer intensity, whereas regions with
lower temperatures tend to have weaker Es layer intensity. The strong correlation between Es layer
intensity and lower atmospheric temperature may be attributed to the influence of temperature
variations in the lower atmosphere [Zhao et al., 2024], which drive atmospheric motion and
generate atmospheric waves. Additionally, more intense atmospheric waves are generated when
the lower atmospheric temperature is higher. These waves gradually propagate from the lower
atmosphere to the height of the Es layer, affecting the formation process of the Es layer. As a result,
the Es layer intensity shows a high consistency with the surface atmospheric temperature. We will
conduct targeted research to further investigate the correlation between Es layer intensity and
surface temperature.
5. 3 Solar cycle variation characteristics
The correlation between Es layer intensity and solar cycles has been a focal point of debate in
the scientific community. Different scholars have drawn contradictory conclusions, including
positive correlation, negative correlation, and no correlation, based on observations from different
stations [Tan et al., 1985; Maksyutin et al., 2005; Closs et al., 1965; Zuo et al., 2006]. Currently,
there are three main viewpoints. One viewpoint suggests that Es layer intensity is independent of



solar activity, implying no significant influence. Another perspective proposes a weak positive
correlation between Es layer intensity and solar activity, implying that variations in solar cycles
may have a slight impact on Es layer intensity. In contrast, there is a third viewpoint suggesting a
weak negative correlation between Es layer intensity and solar activity, indicating that higher solar
activity could potentially lead to a decrease in Es layer intensity. In 1984, Baggaley conducted a
statistical analysis of data from two stations in the southern hemisphere covering three solar
activity cycles. The study concluded that solar activity and the Es layer were not correlated
[Baggaley et al., 1984]. However, the following year, Baggaley found that Es layer intensity
increased with an increase in sunspot numbers [Baggaley et al., 1985].
To investigate the solar cycle variations in Es layer intensity in East Asia, figure 8 utilizes
data from five different latitude observation stations: Manzhouli, Beijing, Chongqing, Guangzhou,
and Haikou. The data used includes the monthly median foEs and sunspot numbers from 1998 to
2020, covering two complete solar cycles. This analysis aims to examine the correlation between
Es layer intensity and solar activity in East Asia.

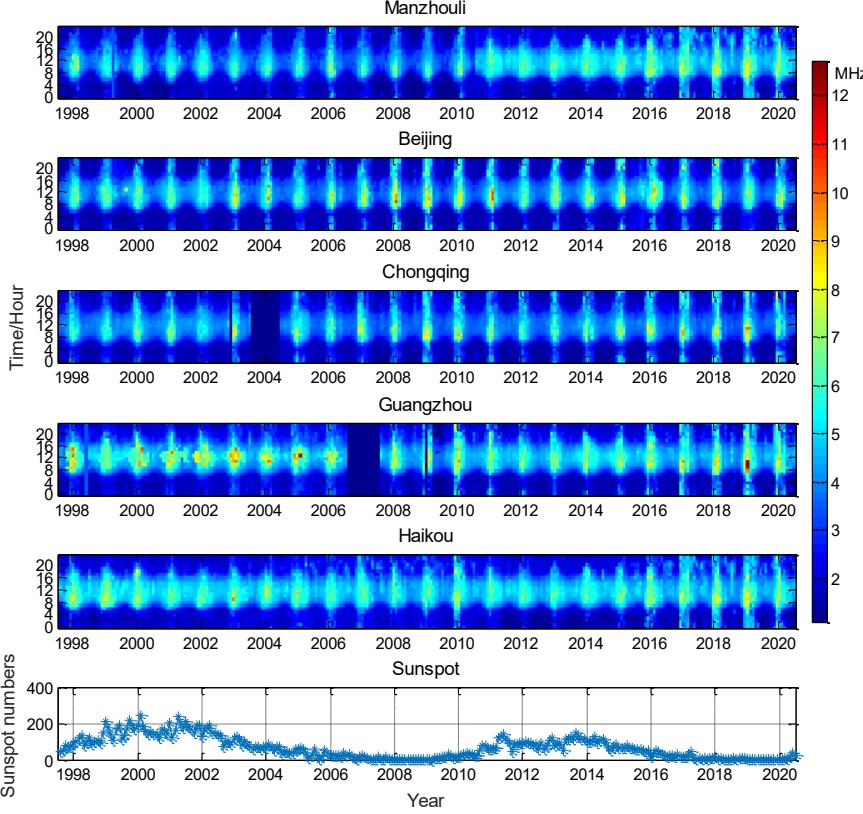


Fig.8 the monthely median values of Es from 1998 to 2020



244   Figure 8 shows that during periods of low solar activity, such as 2006-2009 and 2017-2020,

245 the overall foEs layer intensity is significantly higher compared to periods of low solar activity,

246 such as 1999-2002 and 2011-2014. The Es layer in East Asia exhibits a clear negative correlation

247 with solar activity intensity. Specifically, during years of high solar activity, the nighttime Es layer

248 intensity is notably lower than during years of low solar activity.

249   In order to further investigate the correlation between Es layer intensity and solar activity

250 cycles, the Pearson correlation coefficient was employed to calculate the correlation between

251 daytime and nighttime monthly median foEs values and solar activity. The calculation formulas

252 are as follows:

$$COR(X,Y) = \frac{cov(X,Y)}{\sigma_X \sigma_Y} = \frac{E(XY) - E(X)E(Y)}{\sqrt{E(X^2) - E^2(X)}\sqrt{E(Y^2) - E^2(Y)}} \quad (1)$$

254 where $X$ represents the Es layer critical frequency, and $Y$ represents the number of sunspots.

255 The correlation calculation results between the monthly median foEs and solar activity are

256 presented in Table 2.

257       Table 2 the correlation coefficient between Es layer intensity and solar activity

| Index | Station name | Country | Mean correlation coefficient | Daytime correlation coefficient | Nighttime correlation coefficient |
|---|---|---|---|---|---|
| 1 | Beijing | China | -0.3031 | -0.2665 | -0.4431 |
| 2 | Changchun | China | -0.0198 | 0.1201 | -0.2665 |
| 3 | Chongqing | China | -0.0133 | 0.0724 | -0.0857 |
| 4 | Guangzhou | China | -0.0629 | 0.0327 | -0.0919 |
| 5 | Haikou | China | -0.1295 | 0.0541 | -0.1836 |
| 6 | Lanzhou | China | -0.0664 | 0.1867 | -0.2531 |
| 7 | Lhasa | China | 0.0259 | 0.1494 | -0.1150 |
| 8 | Manzhouli | China | -0.0970 | -0.0194 | -0.2780 |
| 9 | Urumchi | China | -0.0510 | 0.0771 | -0.1510 |
| 10 | Qingdao | China | -0.1138 | -0.0591 | -0.1832 |
| 11 | Sheshan | China | 0.0518 | 0.0469 | 0.0605 |
| 12 | Kunming | China | 0.0363 | 0.0794 | 0.0113 |
| 13 | Xinxiang | China | 0.0858 | 0.1973 | -0.0057 |
| 14 | Suzhou | China | 0.0589 | 0.1536 | -0.0160 |
| 15 | Xian | China | -0.0805 | -0.0001 | -0.1592 |
| 16 | Wuhan | China | 0.1213 | 0.1571 | 0.1682 |
| 17 | Akita | Japan | -0.2905 | -0.3110 | -0.3664 |
| 18 | Okinawa | Japan | -0.3487 | -0.3007 | -0.3910 |
| 19 | Yamagawa | Japan | -0.3321 | -0.3277 | -0.3585 |
| 20 | Wakkanai | Japan | 0.0522 | 0.0847 | 0.0650 |
| 21 | Koku | Japan | 0.0553 | 0.1399 | -0.0485 |
| 22 | Average | – | -0.0657 | 0.0559 | -0.1364 |

258   From Table 2, it is shown that there is an overall negative correlation between foEs in East

259 Asia and sunspot numbers. At daytime, most of the stations exhibit a weak positive correlation

260 between foEs and sunspot numbers. However, during the nighttime, almost all stations show a

261 negative correlation between foEs and sunspot numbers.



5. 4 Long-term variation characteristics
To investigate the long-term variation trends of Es layer intensity in East Asia, figure 9
illustrates the annual variations in monthly median foEs values at eight representative stations:
Manzhouli, Changchun, Urumqi, Beijing, Lanzhou, Chongqing, Haikou, and Guangzhou. The red
line is a linear fit of the monthly median foEs, its expression is:
$$f(x) = bx + a \qquad (2)$$

where $b$ represents the slope, and $a$ represents the constant term.

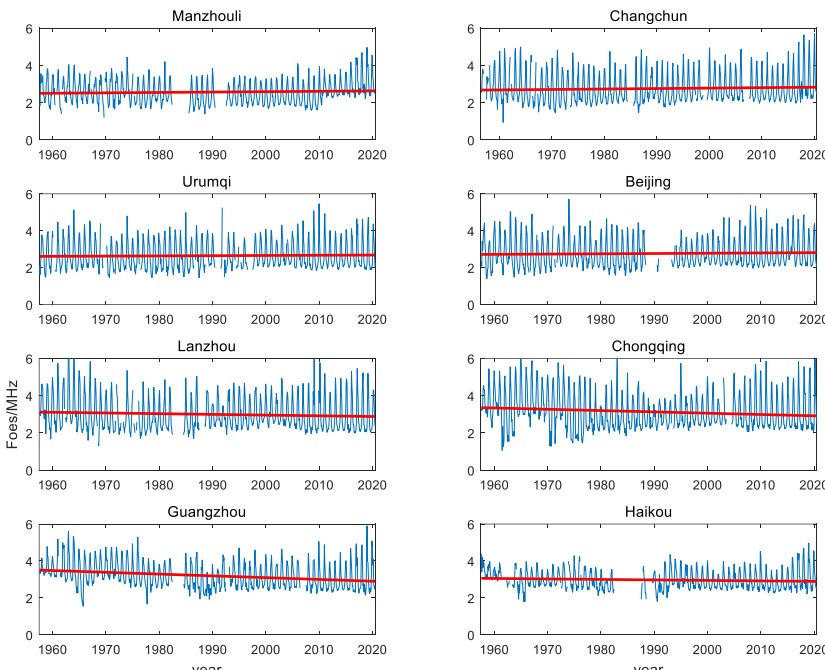


Fig.9 the long-term variation trend of foEs monthly median

From figure 9, it is shown that among the eight stations: Manzhouli, Changchun, Urumqi,
Beijing, Lanzhou, Chongqing, Haikou, and Guangzhou, the stations with higher overall Es layer
intensity exhibit a decreasing trend in monthly median foEs values, while stations with lower Es
layer intensity show an increasing trend. Specifically, the foEs values at Manzhouli, Changchun,
Urumqi, Beijing, and Haikou demonstrate an upward trend, with respective positive slopes of
0.0002, 0.0002, 0.0001, 0.0001, and 0.0002. On the other hand, the Es layer at Lanzhou,
Chongqing, and Guangzhou shows a downward trend, with respective negative slopes of -0.0003,
-0.0006, and -0.0008. By applying the same methodology to study the long-term variation trends
of foEs at 13 other stations (as shown in Table 3), it was found that the Es layer intensity exhibited





a long-term decreasing trend at four stations, with negative slopes ranging from 0 to -0.0010,
while it showed a long-term increasing trend at nine stations, with positive slopes ranging from 0
to 0.0024. Furthermore, the amplitude of Es layer intensity varies across different latitude stations,
with the highest amplitude observed near the 30° latitude line, gradually decreasing towards lower
and higher latitudes.

Table 3 long-term variation trend of Es

| Index | Station name | Country | Slope | Constant term |
|---|---|---|---|---|
| 1 | Beijing | China | 0.0001 | 2.7164 |
| 2 | Changchun | China | 0.0002 | 2.6793 |
| 3 | Chongqing | China | -0.0006 | 3.3586 |
| 4 | Guangzhou | China | -0.0008 | 3.5047 |
| 5 | Haikou | China | 0.0002 | 3.0587 |
| 6 | Lanzhou | China | -0.0003 | 3.118 |
| 7 | Lhasa | China | -0.0005 | 3.2436 |
| 8 | Manzhouli | China | 0.0002 | 2.5047 |
| 9 | Urumchi | China | 0.0001 | 2.6113 |
| 10 | Qingdao | China | 0.0003 | 3.0038 |
| 11 | Sheshan | China | 0.0006 | 3.3815 |
| 12 | Kunming | China | 0.0025 | 2.9697 |
| 13 | Xinxiang | China | 0.0002 | 3.0314 |
| 14 | Suzhou | China | 0.0007 | 3.1915 |
| 15 | Xian | China | 0.0024 | 2.9235 |
| 16 | Wuhan | China | -0.0003 | 3.1730 |
| 17 | Akita | Japan | 0.0003 | 3.1427 |
| 18 | Okinawa | Japan | 0.0001 | 3.0874 |
| 19 | Yamagawa | Japan | 0.0000 | 3.1913 |
| 20 | Wakkanai | Japan | -0.0010 | 3.4483 |
| 21 | Koku | Japan | -0.0009 | 3.5679 |
| 22 | Average | – | 0.00017 | 3.091 |

Analysis of the monthly median foEs values at 21 stations in East Asia reveals an overall
long-term increasing trend in Es layer intensity, with an average positive slope of 0.00017.
Stations with higher Es layer intensity generally exhibit a long-term decreasing trend, while
stations with lower Es layer intensity tend to show a long-term increasing trend. This overall
pattern indicates a negative feedback characteristic. The underlying reasons for the long-term
variation trends in the Es layer could potentially be associated with long-term climate variations.
In our future work, we plan to conduct a more in-depth investigation specifically focused on the
long-term variation trends of the Es layer.
## 6 Conclusion
This study utilizes over 60 years of Es layer observation data from 21 ionospheric vertical
sounding stations in China and Japan to investigate in-depth the characteristics of Es layer
intensity, spatial distribution, diurnal variation, seasonal variation, and long-term trends in East



Asia. The research findings of this study are of significant importance for exploring the causes of
the Es layer, analyzing the spatiotemporal distribution of Es layer intensity. The following
research conclusions have been obtained:
(1) In East Asia, the intensity of foEs during the summer months (May to August) is
significantly higher than in other seasons. Additionally, the intensity is notably higher around local
noon compared to other times of the day. Moreover, the Es layer intensity exhibits strong regional
variations. In general, the maximum intensity of the Es layer is located near the 30° latitude in the
northern Hemisphere, and weakens to lower and higher latitudes. The intensity tends to be higher
in lower latitudes compared to higher latitudes, and the eastern region shows slightly higher
intensity compared to the western region. The monthly average foEs values at all stations have a
maximum value above 5 MHz, with certain stations reaching even above 9 MHz, which is much
higher than the global average level.
(2) At daytime in East Asia, the center of Es layer intensity is observed in the Chongqing,
Guangzhou, and Suzhou areas of China. However, during the nighttime, the center of Es layer
intensity migrates towards the northeast, with the strongest region located in areas such as Suzhou
and Qingdao in China, as well as Koku and Yamagawa in Japan. The diurnal asymmetry of the Es
layer center may be influenced by factors such as the distribution of land and sea, as well as
climatic conditions.
(3) During the summer in East Asia, the center of Es layer intensity is located near 30°N and
exhibits a belt-like distribution. In the winter, the center of Es layer intensity migrates southward
to the Guangzhou and Haikou.
(4) In East Asia, the Es layer intensity in East Asia showed a negative correlation with the
number of sunspots overall, with diurnal inconsistency, weak positive correlation during the day
and negative correlation at night.
(5) Based on the ionosonde data from 21 stations in East Asia, the long-term variation trend
of Es layer intensity at different locations is different, but overall, it presents a long-term upward
trend and has a negative feedback characteristic. The regions with higher average Es layer
intensity showed a long-term downward trend, while the regions with lower average Es layer
intensity showed a long-term upward trend.

**Acknowledgement** Work on this study was supported by the National Natural Science Foundation
of China (No.42074225, 12105251, 52371354, 62201529), National Key Laboratory Foundation
of Electromagnetic Environment (No.6142403240302, 6142403240301), the Stable-Support
Scientific Project of China Research Institute of Radiowave Propagation (No.A132312217-001),
and Stable-Support Scientific Project of Beijing Vacuum Elec-tronics Research Institute under
Grant (No. A240100880). The Es layer data used in the article were all from the National Institute



of Information and Communications Technology (NICT) in Japan. We would like to express our
gratitude.
**Data Availability Statement** The Es data over China and Japan can be available at:
https://github.com/zhaohaisheng22s/-Sporadic-E-Over-East-Asia/commits/Es.
DOI:10.5281/zenodo.10885736.

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

**Author contributions**
Zhao., Feng., Liu. and Xu. wrote the main manuscript text. Xue., Wu., and Xue. prepared Figs. 1
– 6. Peng. and Ding prepared Figs. 7–9. All authors reviewed the manuscript.
**Competing interests**
Te authors declare no competing interests.