# Peer review of "based on long-term data 3 Hai-Sheng Zhao 1, Jie Feng\* 2, Yang Liu 2, Zheng-Wen Xu 2, Jian Wu 3, Kun Xue 2, 4 Huai-Yun Peng2, Zing-Hua Ding3 5 6 1 School of Electronic Science and Technology, Hainan University, Haikou 570228, China 7 2 National Key Laboratory of Electromagnet"

_EGUsphere, 2025_

## Referee Comment (RC1)

The paper "**Variation characteristics of sporadic-E layer in East Asia**" by Zhao et al. deals with a climatological study on the Es occurrence over East Asia based on a significant number of ionosonde stations.

Even though I understand that the authors have done a lot of work to obtain their results, at the same time I think that what they shown is not novel (for instance, "At daytime, foEs values are significantly higher than during the nighttime", "the maximum values of foEs in East Asia generally occur in June, while the minimum values typically occur in December. The foEs values in summer are significantly higher than in winter"). It is unclear what their study brings to the ionospheric community that wasn't unknown before.

-My major concern is related to Section 2 and specifically to the dataset used to perform the analyses. There is no discussion about how the authors have obtained the foEs values. Are we talking about manual or automatic values? What kind of ionosondes they considered (Digisonde, CADI, IPS-42,…..)? What is the time resolution of their data? Does this time resolution change from station to station? The authors should clarify better to the reader this crucial point of their work.

-Why Figure 2 refers to 20 and not to 21 stations? Do these plots refer to the whole dataset of each ionosonde station? Do these plots show hourly monthly medians?

-line 134, "The maximum monthly average…." I am confused, are we talking about medians or averages?

-lines 147 and line 150, Wakkanai and Suzhou should be highlighted somehow in the figure. Moreover, I am pretty doubtful about the application of the Kriging method to the foEs values, also because from the figures I see many spots that in my opinion are unreal. This is a critical point characterizing Figures 3, 5 and 7.

-line 156, what do the authors mean with "the average variations of monthly median foEs values"? The same stands for line 181.

-lines 176-178, "The diurnal drift in the center of Es layer may be affected by environmental factors such as the diurnal variations of background atmosphere and climate." This sentence is too speculative.

-lines 200-201, "However, the occurrence of anomalous phenomena in the mid-latitude Es layer during summer poses a challenge to the wind shear theory." With regard to this, the paper by Haldoupis (2007) that the authors cite partially solves this puzzle.

-lines 207-208, "They also predict that the modulation of tides by planetary waves is achieved through nonlinear interference [Xu et al., 2022]." Concerning this topic, this is not the right reference. Consider: Haldoupis and Pancheva (2002), http://dx.doi.org/10.1029/2001JA000212; Haldoupis et al. (2004), http://dx.doi.org/10.1029/2003JA010253; Pignalberi et al. (2015), http://dx.doi.org/10.1016/j.jastp.2014.10.017; Pezzopane et al. (2015), http://dx.doi.org/10.1016/j.jastp.2015.11.010.

-line 209, what do the authors mean with "probability distribution"?

-line 246, the authors say that "The Es layer in East Asia exhibits a clear negative correlation…" but I wouldn't say that. On the other hand, what the authors write referring to Table 2 highlights that the negative correlation is not as clear as it is claimed here.

-line 254, "number of sunspots", which one? Monthly? Annual?......

-the trends shown in Figure 9 in my opinion cannot be considered statistically significant.

-lines 289-290, what do the authors mean with "This overall pattern indicates a negative feedback characteristic."

**Minor remarks:**

-line 17, it is unclear what is the meaning of "global average". The same concern stands for lines 136 and 309.

-line 32, consider also the paper by Pietrella et al. (2014), http://dx.doi.org/10.1016/j.asr.2014.03.019

-lines 57-59, consider also the following papers: Pignalberi et al. (2014), http://dx.doi.org/10.1016/j.jastp.2014.10.017, Pezzopane et al. (2015), http://dx.doi.org/10.1016/j.jastp.2015.11.010

-delete lines 92-93 from "Additionally….", they are unnecessary.

-delete lines 102-109 from "Particularly….", they are not unnecessary.

-Table 1, switch "longitude" with "latitude".

-line 140, when talking about Kriging, those are not the right references to cite. Consider the following ones: Kitanidis PK (1997) Introduction to geostatistics: application to hydrogeology. Cambridge University Press, Cambridge; Matheron G (1963) Principles of geostatistics. Econ Geol 58:1246–1266; Oliver MA, Webster R (1990) Kriging: a method of interpolation for geographical information systems. Int J Geogr Inf Syst 4(3):313–332.

-line 226, cite also the paper by Pezzopane et al. (2015) which is present in the References section but not cited in the text.

-line 243, replace "monthely" with "monthly".

-line 245, replace "low" with "high".

-lines 352-355, the references are not sorted.

-line 362, this reference is cited in the text as 2006.

-line 399, the reference Pezzopane et al. (2015) is not cited in the text.

---

## Author Comment (AC1)

*Manuscript No.:* egusphere-2025-1524

Manuscript Title: Variation characteristics of sporadic-E layer in East Asia based on

long-term data

**Replies to Reviewer 1's Comments:**

The paper "Variation characteristics of sporadic-E layer in East Asia" by Zhao et al. deals with a climatological study on the Es occurrence over East Asia based on a significant number of ionosonde stations. Even though I understand that the authors have done a lot of work to obtain their results, at the same time I think that what they shown is not novel (for instance, "At daytime, foEs values are significantly higher than during the nighttime", "the maximum values of foEs in East Asia generally occur in June, while the minimum values typically occur in December. The foEs values in summer are significantly higher than in winter"). It is unclear what their study brings to the ionospheric community that wasn't unknown before.

*Reply:* Thank you very much for taking the time to review this manuscript and provide valuable feedback. We greatly appreciate the constructive comments and suggestions, which have helped us improve the quality and clarity of our work.

As you rightly pointed out, while the Es layer in East Asia follows the general variation patterns typical of mid-latitude regions, it also exhibits unique regional characteristics. For instance, the intensity center of the Es layer in East Asia is not fixed but migrates with diurnal and seasonal variations. Although long-term variation trends of Es layer intensity differ across locations in this region, areas with higher intensity generally exhibit a downward trend, while those with lower intensity show an upward tendency. The discovery of these new phenomena provides valuable references for studying Es layer formation mechanisms and atmosphere-ionosphere coupling processes.

(1) My major concern is related to Section 2 and specifically to the dataset used to perform the analyses. There is no discussion about how the authors have obtained the foEs values. Are we talking about manual or automatic values? What kind of ionosondes they considered (Digisonde, CADI, IPS-42,…..)? What is the time resolution of their data? Does this time resolution change from station to station? The authors should clarify better to the reader this crucial point of their work.

*Reply:* Thank you for your comments. All Chinese stations employ the domestically produced CPA-4 ionosonde, while Japanese stations utilize the American-made Digisonde digital ionosonde. The temporal resolution of the data used in this study is 1 hour, with both Chinese and Japanese stations maintaining the same resolution. The

data interpretation method employed is manual analysis. A clarification has been provided in Lines 124-128 of the revised manuscript.

- (3) Why Figure 2 refers to 20 and not to 21 stations? Do these plots refer to the whole dataset of each ionosonde station? Do these plots show hourly monthly medians? *Reply:* Thank you for your comments. Among the 21 stations, Suzhou and Sheshan are geographically proximate. Considering both this spatial overlap and the presentational challenges of displaying 21 subplots, Figure 2 omits the results from Sheshan station. The data presented in the figures and tables of this paper are derived from the complete time periods specified in Table 1, with no data screening applied. Figure 2 displays the annual averages of hourly monthly median values.
- (3) line 134, "The maximum monthly average..." I am confused, are we talking about medians or averages?

*Reply:* Thank you for your comments. We sincerely apologize for the confusion caused by the terminology inaccuracy. The correct term should be "monthly median", not "monthly average". We have corrected this terminology error in Line 143-145 of the original text.

(4) lines 147 and line 150, Wakkanai and Suzhou should be highlighted somehow in the figure. Moreover, I am pretty doubtful about the application of the Kriging method to the foEs values, also because from the figures I see many spots that in my opinion are unreal. This is a critical point characterizing Figures 3, 5 and 7.

**Reply:** Thank you for your comments. We have marked both Sheshan and Suzhou stations in Figure 1. Kriging interpolation is a classical spatial algorithm widely employed in studies of ionospheric and atmospheric distribution characteristics. The apparent distorted patches in the figures are primarily attributable to the rendering effects of the plotting software.

(5) line 156, what do the authors mean with "the average variations of monthly median foEs values"? The same stands for line 181.

*Reply:* Thank you for your comments. Figure 4 displays the monthly median variation curves of typical stations across different months. Each year's data corresponds to one curve, and we applied a multi-year averaging method to the monthly medians to enhance the generalizability of the patterns.

(6) lines 176-178, "The diurnal drift in the center of Es layer may be affected by environmental factors such as the diurnal variations of background atmosphere and climate." This sentence is too speculative.

*Reply:* Thank you for your comments. We have revised this statement as requested. The revised statement reads as follows:

The observed diurnal asymmetries in the intensity of the Es in East Asia may result from variations in the dominant controlling factors of foEs across different periods. During daytime, the electron density of the Es is primarily governed by solar radiation, showing significant latitudinal dependence. However, when solar radiation weakens at night, its controlling effect diminishes, allowing the influence of other factors such as tides and gravity waves to become more pronounced. This may be the cause of the diurnal inconsistency in the Es layer intensity center.

(7) lines 200-201, "However, the occurrence of anomalous phenomena in the mid-latitude Es layer during summer poses a challenge to the wind shear theory." With regard to this, the paper by Haldoupis (2007) that the authors cite partially solves this puzzle.

**Reply:** Thank you for your comments. We sincerely appreciate your insightful perspective on the seasonal dependence mechanism of the Es layer. It should be noted that this explanation represents only one of several plausible hypotheses and has not yet fully resolved the scientific inquiry, primarily for two reasons: first, the correlation between meteor flux and foEs in the literature shows poor fitting across multiple time periods; second, the scientific validity of directly equating meteor counts with foEs requires further scrutiny.

(8) lines 207-208, "They also predict that the modulation of tides by planetary waves is achieved through nonlinear interference [Xu et al., 2022]." Concerning this topic, this is not the right reference. Consider: Haldoupis and Pancheva (2002),Haldoupis al. http://dx.doi.org/10.1029/2001JA000212; et (2004),Pignalberi http://dx.doi.org/10.1029/2003JA010253; al. (2015),et (2015),http://dx.doi.org/10.1016/j.jastp.2014.10.017; Pezzopane et al. http://dx.doi.org/10.1016/j.jastp.2015.11.010.

**Reply:** Thank you for your comments. We sincerely appreciate your reminder and have corrected the relevant references in the revised manuscript.

(9) line 209, what do the authors mean with "probability distribution"? *Reply:* Thank you for your comments. We sincerely apologize for the confusion caused by unclear expression. The term "probability distribution" here specifically refers to the occurrence probability of foEs values exceeding 5 MHz. We have corrected this statement in Line 223 of the revised manuscript..

(10) line 246, the authors say that "The Es layer in East Asia exhibits a clear negative correlation ..." but I wouldn't say that. On the other hand, what the authors write referring to Table 2 highlights that the negative correlation is not as clear as it is claimed here.

**Reply:** Thank you for your comments. We sincerely apologize for the confusion caused by the unclear expression. The analysis results indicate that foEs exhibits a weak correlation with sunspot numbers, rather than a significant one. We have revised this statement accordingly in the revised manuscript line 258-260.

(11) line 254, "number of sunspots", which one? Monthly? Annual?.....

*Reply:* Thank you for your comments. The sunspot number data used in this study are monthly mean values. A clarification on this issue has been provided in line 266 of the revised manuscript.

(12) the trends shown in Figure 9 in my opinion cannot be considered statistically significant.

**Reply:** Thank you for your comments. We concur with your perspective. While environmental changes exhibit long-term characteristics, and the observation period covered in this study may not be sufficient to fully capture the long-term variation patterns of foEs, we present these statistically observed phenomena as data-driven findings for readers' reference.

(13) lines 289-290, what do the authors mean with "This overall pattern indicates a negative feedback characteristic."

**Reply:** Thank you for your comments. Statistical results indicate that regions with high foEs intensity exhibit a long-term downward trend, while areas with low intensity show a long-term upward trend. This pattern is analogous to the "negative feedback circuit" principle in electronics, hence we describe it as possessing negative feedback characteristics.

**Minor remarks:**

(14) line 17, it is unclear what is the meaning of "global average". The same concern stands for lines 136 and 309.

*Reply:* Thank you for your comments. The term "global average" here refers to the average value level of foEs on a global scale.

(15) Line 32, consider also the paper by Pietrella et al. (2014), http://dx.doi.org/10.1016/j.asr.2014.03.019

*Reply:* Thank you for your comments. We have added this reference as requested in the revised manuscript.

(16) lines 57-59, consider also the following papers: Pignalberi et al. (2014), http://dx.doi.org/10.1016/j.jastp.2014.10.017, Pezzopane et al. (2015), http://dx.doi.org/10.1016/j.jastp.2015.11.010

**Reply:** Thank you for your comments. We have added citations to these two references at the appropriate locations in the revised manuscript.

(17) delete lines 92-93 from "Additionally....", they are unnecessary.

*Reply:* Thank you for your comments. We have removed this statement from the revised manuscript.

(18) delete lines 102-109 from "Particularly".", they are not unnecessary.

**Reply:** Thank you for your comments. We have removed this statement from the revised manuscript.

(19) Table 1, switch "longitude" with "latitude".

*Reply:* Thank you for your reminder. We have completed the corresponding revisions in the revised manuscript.

(20) line 140, when talking about Kriging, those are not the right references to cite. Consider the following ones: Kitanidis PK (1997) Introduction to geostatistics: application to hydrogeology. Cambridge University Press, Cambridge; Matheron G (1963) Principles of geostatistics. Econ Geol 58:1246 – 1266; Oliver MA, Webster R (1990) Kriging: a method of interpolation for geographical information systems. Int J Geogr Inf Syst 4(3):313 – 332.

*Reply:* Thank you for your comments. We have replaced the relevant references in the revised manuscript.

(21) line 226, cite also the paper by Pezzopane et al. (2015) which is present in the References section but not cited in the text.

*Reply:* Thank you for your comments. We have added a citation to this reference in the revised manuscript.

(22) Table 1, switch "longitude" with "latitude".

**Reply:** Thank you for your reminder. We have completed the corresponding revisions in the revised manuscript.

(23) line 243, replace "monthely" with "monthly".

*Reply:* Thank you for your reminder. We have made the corresponding corrections in the revised manuscript.

(24) line 245, replace "low" with "high".

*Reply:* Thank you for your reminder. We have made the corresponding corrections in the revised manuscript.

(25) lines 352-355, the references are not sorted

*Reply:* Thank you for your reminder. We have made the corresponding corrections in the revised manuscript.

(26) line 362, this reference is cited in the text as 2006.

*Reply:* Thank you for your reminder. We have made the corresponding corrections in the revised manuscript.

(27) line 399, the reference Pezzopane et al. (2015) is not cited in the text.

*Reply:* Thank you for your reminder. We have added a citation to this reference in the revised manuscript.

Thank you for your great effort and valuable time spent in reviewing this paper. We sincerely wish that with the careful revision of the paper, the revised manuscript is acceptable for publication.

---

## Author Comment (AC2)

*Manuscript No.:* egusphere-2025-1524

Manuscript Title: Variation characteristics of sporadic-E layer in East Asia based on

long-term data

**Replies to Reviewer 1's Comments:**

Maybe it's my fault but I couldn't download the revised version of the paper. I didn't find a link from which to download it.

Overall, I think that the answers given by the authors are pretty fair and acceptable, except the one related to the Kriging method.

**Reply:** Thank you very much for taking the time to review this manuscript and provide valuable feedback. We greatly appreciate the constructive comments and suggestions, which have helped us improve the quality and clarity of our work. I sincerely apologize, but for some reason, the submission system is preventing me from uploading the revised manuscript.

When the authors say that "Kriging interpolation is a classical spatial algorithm widely employed in studies of ionospheric and atmospheric distribution characteristics. The apparent distorted patches in the figures are primarily attributable to the rendering effects of the plotting software." I agree that "Kriging interpolation is a classical spatial algorithm widely employed in studies of ionospheric and atmospheric distribution characteristics. " but soon after when the authors say "The apparent distorted patches in the figures are prim0arily attributable to the rendering effects of the plotting software." cannot be accepted. What some figures in the paper are showing is not real, it is an artifact. In general, the first thing that should be evaluated is whether there exist the right conditions to apply the Kriging method, but in general, to apply any method to interpolate a discrete dataset of points. In my opinion, the information given but some figures shown by the authors is a little bit distorted.

Reply: Thank you for your comments. During the Kriging interpolation process, grid discretization of the interpolation area inevitably introduces errors. Therefore, the appearance of minor distorted structures in interpolated images is unavoidable. However, it can be confirmed that these distortions are not attributable to data issues. The degree of image distortion is significantly influenced by the chosen standard color map. The figure below demonstrates the visual effects of applying both 'jet' and 'parula' color maps to Figure 3 from this paper. As can be observed, the distortion presented by the 'parula' color map is markedly less pronounced than that of the 'jet' color map. In the revised manuscript, we have standardized the use of the 'parula' color map for all Kriging interpolation results, which has helped mitigate the image distortion to some extent.

Fig.3 the distribution of foEs average values in East Asia (jet)

Fig.3 the distribution of foEs average values in East Asia (parula)

Thank you for your great effort and valuable time spent in reviewing this paper. We sincerely wish that with the careful revision of the paper, the revised manuscript is acceptable for publication.

---

## Author Comment (AC3)

*Manuscript No.:*  egusphere-2025-1524
*Manuscript Title:* Variation characteristics of sporadic-E layer in East Asia based on long-term data

**Replies to Reviewer 2's Comments:**

This manuscript analyzes the morphology of sporadic E (Es) layers over the Asian region using foEs measurements from 21 ground-based ionosonde stations. While the topic is relevant and the dataset has the potential to contribute to regional Es climatology, the main conclusions presented in the manuscript are not sufficiently supported by rigorous analysis. Substantial revisions would be required before the study could be considered for publication. My detailed comments are provided below.

*Reply:* Thank you very much for taking the time to review this manuscript and provide valuable feedback. We greatly appreciate the constructive comments and suggestions, which have helped us improve the quality and clarity of our work.

(1) Line 112-117. The paragraph does not clearly explain how the raw foEs data were preprocessed. Key information is missing, including a) the altitude range or criteria used to select valid Es, b) how missing data were identified and removed, c) the total volume of raw data and processed data. Without these details, the reproducibility and reliability of the results cannot be evaluated.

*Reply:* Thank you for your comments. In this study, the method for selecting valid Es is as follows: the virtual height of the echo must fall within the range of 90–140 km, and the trace should exhibit a horizontally thin-layer structure (traces showing parabolic shapes are generally identified as regular E-layer echoes and thus excluded). During data processing, missing data are labeled as "NaN" and automatically excluded from statistical analyses.

The original data have a sampling interval of one hour, yielding 24 foEs values per day. The total amount of raw data for one year is approximately $24 \times 30$ (days per month) $\times$ 12 (months). After processing, only the monthly median values are retained; thus, the total amount of processed data for one year is reduced to $24 \times 12$ (i.e., one median foEs value per hour over 12 months).

(2) Inconsistent use of median vs. mean. The manuscript should justify why the median was chosen instead of the mean or maximum value for characterizing foEs. For example, in Figure 2, the author used "median mean", but it is unclear how it was calculated. In Figure 3, the "annual mean" is used instead of the median. Why do authors need to manually select different metrics?

*Reply:* Thank you for your comments. We sincerely apologize for the confusion caused by the authors' unclear expressions, which may have hindered the reviewer's understanding of the paper. The foEs dataset selected for this study includes a large number of stations, covers a long time span, and contains a substantial volume of data. To facilitate the investigation of the long-term variation characteristics of foEs over East Asia, we adopted the monthly median at each station as the basic research unit, rather than focusing excessively on data from specific individual days.

During statistical analysis of these monthly medians—such as examining their diurnal variation, seasonal variation, annual variation, and solar cycle variation—it is necessary to take averages of the monthly medians. This has led to situations in the text where median, mean, and even the term "median mean" appear in different places, depending on the specific statistical context.

(3) Line 244-245. The manuscript claims that foEs values during low solar activity are stronger than during high solar activity. However, Figure 8 only displays colorbar without specific values, making the conclusion unverifiable. Before drawing such conclusions, quantitative comparisons, such as the average change or spectral changes, must be made.

*Reply:* Thank you for your comments. To conduct a quantitative comparison, we selected the years 1999–2002 and 2011–2014 as representative solar maximum periods (years of high solar activity), and 2006–2009 and 2017–2020 as solar minimum periods (years of low solar activity). We then compared the overall annual mean foEs values for these periods. The results show that the average foEs during solar minimum years is higher than during solar maximum years, with an increase of approximately 0.1–0.3 MHz. The foEs values for representative stations are listed in Table 1.

Table 1 Comparison of the average foEs values between solar maximum years and solar minimum years

| Index | Station name | Average foEs value in solar maximum 1999-2002 (MHz) | Average foEs value in solar maximum 2011-2014 (MHz) | Average foEs value in solar minimum 2006-2009 (MHz) | Average foEs value in solar minimum 2017-2020 (MHz) |
|---|---|---|---|---|---|
| 1 | Manzhouli | 2.3870 | 2.7498 | 2.7278 | 3.0862 |
| 2 | Beijing | 2.7574 | 2.7954 | 2.8460 | 2.8793 |
| 3 | Chongqing | 3.0522 | 3.0016 | 3.0953 | 3.1690 |
| 4 | Guangzhou | 3.0612 | 2.9079 | 3.0062 | 3.1764 |
| 5 | Haikou | 2.8560 | 2.7498 | 2.9502 | 3.1858 |

(4) Previous studies have shown that weak foEs (< 3 MHz) are more sensitive to solar cycle modulation (see Figure 6 in Tian et al., 2024. Check https://doi.org/10.1029/2024JH000279). Table 2 in the present manuscript shows

correlations between overall foEs and sunspot number, but it does not distinguish between weak and strong foEs. I encourage the authors to examine the correlations between weak/strong foEs and solar activity. Such an analysis would help clarify whether the solar cycle dependence of Es morphology is driven primarily by weak events, strong events, or both.

*Reply:* Thank you for your comments. To further analyze the correlations between weak/strong foEs and solar activity, Table 2 presents the calculated occurrence probabilities of weak Es (foEs < 3 MHz) and strong Es (foEs > 5 MHz) during solar maximum periods (1999–2002 and 2011–2014) and solar minimum periods (2006–2009 and 2017–2020).

Table 2 Comparison of the occurrence probabilities of weak foEs and strong foEs during years of high and low solar activity

| Index | Station name | Occurrence probability (foEs<3MHz) in solar maximum | Occurrence probability (foEs>5MHz) in solar maximum | Occurrence probability (foEs<3MHz) in solar minimum | Occurrence probability (foEs>5MHz) in solar minimum |
|---|---|---|---|---|---|
| 1 | Manzhouli | 0.4783 | 0.0113 | 0.4284 | 0.0421 |
| 2 | Beijing | 0.5846 | 0.0360 | 0.6102 | 0.0595 |
| 3 | Chongqing | 0.4566 | 0.0807 | 0.4154 | 0.1181 |
| 4 | Guangzhou | 0.4358 | 0.0877 | 0.3364 | 0.0694 |
| 5 | Haikou | 0.4974 | 0.0321 | 0.4488 | 0.0451 |
| 6 | Changchun | 0.5664 | 0.0408 | 0.5204 | 0.0747 |
| 7 | Lanzhou | 0.4536 | 0.0703 | 0.4905 | 0.0777 |
| 8 | Lhasa | 0.4553 | 0.0490 | 0.4271 | 0.0634 |
| 9 | Urumchi | 0.5781 | 0.0204 | 0.5378 | 0.0425 |

The results indicate that the occurrence rate of weak Es is slightly higher during solar maximum than during solar minimum, whereas the occurrence rate of strong Es is significantly higher during solar minimum compared to solar maximum. The occurrence rate of weak Es is slightly higher during solar maximum than during solar minimum, which is consistent with the findings of Tian et al.

Penghao Tian, Bingkun Yu, Hailun Ye, Xianghui Xue, Jianfei Wu, Tingdi Chen, Deep Learning Insights Into Ionospheric Sporadic E Irregularities Under Different Solar Activity Conditions, J. Geophys. Res., 2024, 129, e2024JH000279.

(5) Although the manuscript maps the spatiotemporal distribution of foEs, it provides little discussion of the underlying physical mechanisms. For example, why does the nighttime Es intensity center shift northeastward? Without physical interpretation, the results remain descriptive rather than scientific, limiting the manuscript's contribution to the field.

*Reply:* Thank you for your comments. We have added discussions and analyses of the underlying physical processes and mechanisms in the revised version of the manuscript. Regarding the reason for the northeastward shift of the nighttime Es intensity center, there is currently no definitive conclusion. One possible explanation is that the dominant controlling factors for Es layer formation differ between daytime and nighttime, leading to a shift in the location of the Es intensity center from day to night.

The observed diurnal asymmetries in the intensity of the Es in East Asia may result from variations in the dominant controlling factors of foEs across different periods. During daytime, the electron density of the Es is primarily governed by solar radiation, showing significant latitudinal dependence. However, when solar radiation weakens at night, its controlling effect diminishes, allowing the influence of other factors such as tides and gravity waves to become more pronounced. This may be the cause of the diurnal inconsistency in the Es layer intensity center.

(6) Line 316. One of the main conclusions is that the Es intensity center is located near 30N. However, this conclusion appears to be based solely on the spatial distribution in Figure 2. In the other word, different interpolation methods may lead to different conclusions. To validate this conclusion, satellite observations (e.g., COSMIC-1/2) should be considered. Otherwise, the latitudinal maximum remains uncertain and may be an artifact of the interpolation procedure.

*Reply:* Thank you very much for the reviewer's constructive comments. The ionosonde network in East Asia is relatively sparse, with adjacent stations spaced 3–6 degrees apart in latitude. As a result, the foEs distribution maps obtained through interpolation may indeed contain certain errors. To further validate the conclusion, we plan to incorporate occultation observation data in future work and conduct comparative validation studies between ionosonde and satellite observations.

Thank you for your great effort and valuable time spent in reviewing this paper. We sincerely wish that with the careful revision of the paper, the revised manuscript is acceptable for publication.